# The Association between Modifiable Lifestyle Behaviors and Depression among Asian Americans with Chronic Hepatitis B by Medication Status

**DOI:** 10.3390/brainsci12020188

**Published:** 2022-01-30

**Authors:** Lin Zhu, Wenyue Lu, Winterlyn Gamoso, Yin Tan, Cicely Johnson, Grace X. Ma

**Affiliations:** 1Center for Asian Health, Lewis Katz School of Medicine, Temple University, Philadelphia, PA 19140, USA; lin.zhu@temple.edu (L.Z.); wenyue.lu@temple.edu (W.L.); WINTERLYN.GAMOSO55@myhunter.cuny.edu (W.G.); ytan@temple.edu (Y.T.); 2Department of Urban Health and Population Science, Lewis Katz School of Medicine, Temple University, Philadelphia, PA 19140, USA; 3School of Urban Public Health, Hunter College, City University of New York, New York, NY 10065, USA; 4Hunter College Center for Cancer Health Disparities Research (CCHDR), Hunter College, City University of New York, New York, NY 10065, USA; cjpolicydr@gmail.com

**Keywords:** chronic hepatitis B, depression, lifestyle behaviors, antiviral medication

## Abstract

Asian Americans are disproportionately affected by chronic hepatitis B (CHB), with incidence and mortality rates well above those experienced by non-Hispanic white populations. The goal of this study was to examine the association between depression and modifiable lifestyle behaviors among Asian Americans with CHB, with a comparison between those on hepatitis medication and those not on medication. In total, 313 Asian Americans with CHB were recruited through outpatient clinics and community-based organizations to participate in an in-person baseline assessment. We collected data on participants’ sociodemographic characteristics, health-related behaviors, depression symptoms, and modifiable lifestyle behaviors. Bivariate analyses (two sample t-test and chi-square test of independence) and multivariable logistic regression were conducted. We found a high prevalence of depression among individuals living with CHB (41.81% among those not on antiviral medication and 39.71% among those on medication). Multivariate logistic regression results showed that Chinese ethnicity (vs. Vietnamese) and lack of physical activity were significantly associated with a higher risk of mild/severe depression, regardless of medication status. However, the protective effect of physical activity was strong for those on antiviral medication. Furthermore, being employed was significantly associated with a lower risk for depression among Asian Americans on medication, while younger age and being currently married were significantly associated with lower risk of depression among those not on medication. Our findings highlight the significance of physical activity among Asian Americans with CHB, especially for those on antiviral medication. Future prospective research efforts are needed to better identify the potential behavioral mechanisms of depression and provide insights for the psychopharmacological management in this vulnerable population.

## 1. Introduction

Hepatocellular carcinoma (HCC) is the most common primary liver malignancy, occurring most often in people with chronic liver diseases, such as cirrhosis caused by hepatitis B virus (HBV) or hepatitis C virus (HCV) infection [1]. Globally, 78% of HCC is attributable to HBV (53%) or HCV (25%) [2]. In the United States, Asian Americans are disproportionately affected by liver and intrahepatic bile duct cancer [3]. Asian American men have an 80% higher incidence rate of liver cancer compared to non-Hispanic white men (19.9 vs. 10.8 per 100,000), while Asian American women have twice the incidence rate compared to non-Hispanic white women (7.4 vs. 3.7 per 100,000) [3]. Overall, Asian Americans are 70% more likely to die from liver cancer than their non-Hispanic white counterparts [3]. The mortality rates were particularly high in Vietnamese (54.3), Korean (33.9), and Chinese (23.3) men in the US [3]. Therefore, HCC is considered to be the most important cancer health disparity affecting Asian Americans [4,5].

HBV infection is the leading cause of liver cirrhosis and HCC [6,7,8,9,10]. About 2.2 million individuals in the US are living with chronic HBV (CHB) infection, about 58% of whom are foreign-born Asian Americans [11]. Prevalence rates of CHB in Asian Americans range from 8% to 13%, a figure that is significantly higher than the 1% rate observed in non-Hispanic whites [12,13]. However, CHB management has remained suboptimal in Asian American communities, with low levels of knowledge, awareness, and adherence to regular monitoring or medication treatment [14,15,16,17,18,19,20,21,22,23,24,25], despite increasing efforts in public health campaigns.

One area that is severely understudied is the psychological burden of CHB infection among Asian Americans. Research has found elevated levels of depression and other mood or anxiety disorders among people with CHB and related liver diseases [26,27,28,29,30,31,32,33,34,35,36,37]. Possible explanations for increased depression among patients with CHB include fear of disease progression, fear of transmitting the disease, fear of death, fear of social ostracism or stigma, fear of job loss, restrictions in sexual activity, social isolation, and presence of comorbidities, such as hypertension, osteoporosis, or diabetes mellitus [26,27,28,29,30,31,32,33,35].

So far, researchers have examined several factors associated with depression in CHB patients. Previous studies have suggested that medications may play a significant role in depression and other mood-related disorders and may influence overall quality of life among patients [38,39,40,41,42,43,44,45]. One of the most significant neuro-psychiatric side effects of interferon-alpha (IFN-α), the most widely used treatment for viral hepatitis, is the induction of major depression and suicidal thoughts [40,41]. For example, a study of 32 hepatitis C patients found that IFN-α treatment was associated with significant motor slowing, which was in turn related to the development of depressive symptoms [46]. However, most research has focused on patients with hepatitis C. Of the few studies on patients with CHB, one study found a significant increase in depressive symptoms in HBV patients since beginning treatment, albeit at a lower rate than HCV patients [47]. Another study found a higher prevalence of depression among HBV and HCV patients on IFN therapy than those not on medications, but they did not examine any differences between patients with the two infections [44]. To the best of our knowledge, no studies have specifically assessed the depression burden and related factors among Asian Americans with CHB.

Furthermore, knowledge on the psychosocial and behavioral correlates of depression among Asian Americans living with CHB is limited. Several studies have suggested that sociodemographic factors and lifestyle behaviors might be related to depression among people with HBV or other chronic liver diseases, including older age [48], female gender [29,47], lower income [48], unemployment, excessive drinking [27,36], smoking [36], and presence of comorbidities [33,48]. However, no studies have examined the correlates in Asian Americans with CHB, a population with a heterogenous sociodemographic and behavioral profile. The association between these modifiable lifestyle behaviors and depression could shed light on depression prevention and treatment for Asian Americans with CHB.

To address limitations in the existing literature, we used the baseline data of a randomized controlled clinical trial to evaluate the behavioral and psychological profile of Asian Americans with CHB. The goal of this study was to examine the association between depression and modifiable lifestyle behaviors, including physical activity, smoking, and alcohol use, among Asian Americans with CHB, with separate analysis of those on hepatitis medication and those not on medication.

## 2. Materials and Methods

### 2.1. Study Design

This study utilized baseline data from an ongoing randomized controlled clinical trial aimed at improving long-term adherence to CHB monitoring and treatment. In collaboration with healthcare providers and community-based organizations, we recruited 382 Asian Americans (298 Chinese Americans and 84 Vietnamese Americans) with CHB in the Greater Philadelphia Area and New York City. Participants were eligible to participate in the study if they: (1) were 18 or older; (2) self-identified as of Chinese or Vietnamese descent; (3) had chronic HBV infection with positive HBV surface antigen (HBsAg); (4) were diagnosed at least 12 months ago; (5) were non-compliant to HBV monitoring and treatment guidelines for more than 6 months; (6) were accessible through cellphone to receive text messaging; and (7) were not enrolled in any other HBV management intervention. After listwise deletion, the analysis sample consisted of 313 individuals, including 244 Chinese Americans and 69 Vietnamese Americans. The excluded cases did not differ significantly from the analysis sample on any sociodemographic characteristics except for age, with the average age of the excluded cases being 50 and that of the analysis sample being 53. The subsample from the Greater Philadelphia Area did not differ significantly from that from New York City.

### 2.2. Data Collection

From April 2019 to March 2020, we conducted face-to-face surveys to collect baseline data. Participants were asked about their sociodemographic characteristics, health-related behaviors, depressive symptoms, and modifiable lifestyle behaviors. The questionnaire was designed in English and then translated into Chinese and Vietnamese. The survey was approximately 30 min in length. Bilingual community health educators were on-site to provide language support and other assistance during the survey. The study was approved by the Western Institutional Review Board (WIRB) (protocol No: 20190122). All participants read and signed the informed consent forms to participate in the study.

### 2.3. Measurement

#### 2.3.1. Primary Outcome

The primary outcome of interest for this study was depression severity, assessed with the Patient Health Questionnaire-9 (PHQ-9), a commonly used tool for depression screening in at-risk populations [49]. The PHQ-9 module scores each of the nine DSM-IV criteria as “0” (not at all) to “3” (nearly every day). We added up the responses to all 9 questions to produce a total score, with a possible range from 0 to 27. We then categorized the score into 5 levels of depression severity as follows: minimal depression (0–4), mild depression (5–9), moderate depression (10–14), moderately severe depression (15–19), and severe depression (20–27). We then collapsed the 5 levels into 2 categories to create a dichotomous variable (minimal or no depression vs. mild-severe depression).

#### 2.3.2. Modifiable Lifestyle Behaviors

We collected data on study participants’ modifiable lifestyle behaviors, including smoking, physical activity, and alcohol consumption. Smoking was a dichotomous variable (yes or no). Physical activity was assessed with the question “during the past month, other than your regular job, how many hours do you participate on average in any physical activities or exercises such as running, golf, gardening or walking?”. Those who answered “none” were categorized as “inactive”, the rest as “active”. Alcohol consumption was assessed with the question “how many glasses of alcohol do you drink each week?”. Those who answered “none” were categorized as “non-drinker”, the rest as “drinker”.

#### 2.3.3. Sociodemographic Measures

We also collected information on study participants’ sociodemographics, such as age in years, gender (male or female), ethnicity (Chinese or Vietnamese), number of years living in the US, marital status (currently married/cohabitating or other), educational attainment (≤high school degree or ≥college), employment status (employed, unemployed, or not in labor force), income (<USD 20,000 or ≥USD 20,000), and English-speaking proficiency (not at all/not well or well/very well).

### 2.4. Statistical Analysis

We used chi-square tests to compare the psychosocial, modifiable lifestyle factors, and depression severity by medication status. We conducted binary logistic regressions to examine the effect of modifiable lifestyle behaviors and psychosocial factors on depression severity stratified by medication status. A *p*-value smaller than 0.05 was considered statistically significant. All analyses were conducted in Stata 16 [50].

## 3. Results

Table 1 shows the sociodemographic characteristics of our sample by medication status. Of the 313 total participants, 177 were on antiviral medication and 136 were not on medication. Bivariate analyses revealed significant differences in gender, education, and health insurance coverage by medication status. Specifically, among those on medication, there were significantly higher proportions of male participants (57.35% vs. 41.81%), of participants with a high school or lower degree (74.26% vs. 62.15%), and of participants with health insurance coverage (92.65% vs. 81.92%), compared to the non-medication group. Other sociodemographic characteristics did not differ significantly by medication status. Overall, the study sample consisted of all foreign-born immigrants with low household income and limited English proficiency.

We also examined modifiable lifestyle behaviors, depression, and CHB history among study participants according to antiviral medication status (Table 2). The two subgroups varied significantly only by years since HBV diagnosis, with the average number of years being lower among those on medication (19.08 years) than those not on medication (21.72 years). Overall, the majority of participants did not smoke or drink alcohol at the time. We also found low levels of physical activity, with about one-third of participants, regardless of medication status, reporting that they did not engage in any physical activity. Furthermore, almost 40% of participants had mild or severe depression.

We conducted binary logistic regression to examine how sociodemographic characteristics, HBV history, and modifiable lifestyle behaviors were associated with depression separately among those on CHB mediation and those not on medication (Table 3). Two factors were significant predictors of depression in both groups. Specifically, Chinese participants were more likely than their Vietnamese counterparts to have mild to severe depression, both among those on medication (OR = 7.12, *p* < 0.01) and those not on medication (OR = 8.02, *p* < 0.05), with other variables held constant. Those who reported that they had some physical activity were less likely than those reporting no physical activity at all to have mild to severe depression, both among those on medication (OR = 0.29, *p* < 0.01) and those not on medication (OR = 0.40, *p* < 0.01), with other variables held constant. Additionally, we found that older age (OR = 1.07, *p* < 0.01) was associated with a higher risk for mild or severe depression, while being currently married (OR = 0.30, *p* < 0.05) was associated with a lower risk for depression among those not on antiviral medication. On the other hand, among those on antiviral medication, being unemployed (OR = 4.53, *p* < 0.05) or not in the labor force (OR = 3.44, *p* < 0.05) were significant predictors of a higher risk for mild or severe depression.

## 4. Discussion

In this study, we examined associations between depression and sociodemographic characteristics and modifiable lifestyle behaviors among CHB patients on medication and CHB patients not on medication. There are four main findings. First, we confirmed the previously reported high prevalence of depression among individuals living with CHB [26,32,48], with approximately 40% of participants having mild to severe depression. Previous studies have suggested that Asian Americans with viral hepatitis were more likely than their counterparts of other racial/ethnic groups to experience depression [27]. Therefore, active collaboration between psychiatrists and hepatologists is of particular importance in the treatment of Asian Americans with CHB. However, we did not find elevated depression prevalence among those on antiviral medication, contradicting findings from previous studies [26,27,28,29,30,31,32,33,34,35,36,37]. Given the cross-sectional nature of this study, it is possible that medication-induced depression led to changes in medication status, such as discontinuation of the offending medication [51,52]. A few studies have examined the relationship between medication-induced depression and adherence to medication among patients with viral hepatitis. Although their data showed clear causal evidence for such associations, the researchers did point out the significance of interdisciplinary care and antidepressant treatment as part of the psychopharmacological management in successful treatment of psychiatric symptoms induced by IFN-α [53,54,55,56]. Longitudinal research is needed to further examine the link between medication and depression among Chinese American patients and other Asian ethnic groups.

Second, we found that Chinese Americans with CHB were 7–8 times more likely to suffer depression compared to their Vietnamese counterparts, regardless of medication status, even with control for sociodemographic factors and modifiable lifestyle behaviors. This is, to the best of our knowledge, the first study to examine disparities in depression between these two Asian ethnic groups, both of which are disproportionally affected by CHB. Our finding highlights the need for more research on the predisposition to depression in Chinese Americans living with CHB and on culturally tailored approaches to improving mental health status in both communities.

Third, physical activity was found to be a consistent and significant predictor of lower depression risk, regardless of medication status, which was consistent with the broader body of literature on the protective effects of physical activity on mental health among patients with chronic health conditions [57,58,59,60]. This finding lends support for physical activity interventions that aim to promote exercise in Asian Americans living with CHB. It is noteworthy that the effect of physical activity was strong among those on medication. In other words, for Asian Americans on medication, any physical activity would result in an even greater reduction in depression risk. It is possible that physical exercise could serve as a buffer for medication-induced depression among Asian Americans with CHB. This hypothesis needs to be tested in future empirical studies.

Finally, we identified different sociodemographic predictors of depression between those on medication and those not on medication. Being employed was significantly associated with a lower risk for depression among Asian Americans on medication, whereas younger age and being currently married were significantly associated with a lower risk of depression among those not on medication. This finding shed light on specific demographic groups that may be particularly vulnerable to depression. With better understanding of the mechanisms of such vulnerability, healthcare providers and public health interventions would be able to provide more targeted psychopharmacological management or treatment to prevent and treat depression among patients with CHB.

This study is not without limitations. First, this observational study used cross-sectional data. Therefore, our findings did not indicate causality. To better understand causal mechanisms underlying associations between antiviral medication, physical activity, and depression, prospective studies are needed. Second, researchers have previously suggested that the use of antidepressants is effective in treating IFN-induced depression in patients with viral hepatitis [55,61,62]. However, because we did not collect data on antidepressant use, we were unable to examine the influence of antidepressant use on adherence to antiviral medication. Furthermore, we did not report the level of medication adherence among those on medication in this particular study; however, this will be presented in a subsequent publication.

To the best of our knowledge, this is the first study to examine the prevalence and covariates of depression by medication status among Asian Americans with CHB. Our findings provide important epidemiological estimates for clinical practice and public health interventions and, at the same time, offer new insight into potential behavioral mechanisms of depression, warranting future prospective research.

## Figures and Tables

**Table 1 brainsci-12-00188-t001:** Participant Sociodemographic Characteristics by Medication Status (*n* = 313).

	Not on Medication (*n* = 177)	On Medication (*n* = 136)	*p*-Value
Characteristics	*n* (%) or mean (sd)	*n* (%) or mean (sd)	
Ethnicity			0.78
Chinese	139 (78.53%)	105 (77.21%)	
Vietnamese	38 (21.47%)	31 (22.79%)	
Age	53.68 (13.42)	534.08 (13.05)	0.79
Gender			0.006
Male	74 (41.81%)	78 (57.35%)	
Female	103 (58.19%)	58 (42.65%)	
US-Born	177 (100%)	136 (100%)	
Years living in the US	20.57 (9.64)	20.13 (9.81)	0.69
Marital status			0.06
Currently married	139 (78.53%)	118 (86.76%)	
Other	38 (21.47%)	18 (13.24%)	
Education level			0.02
≤high school	110 (62.15%)	101 (74.26%)	
≥college	67 (37.85%)	35 (25.74%)	
Employment status			0.94
Employed	109 (61.58%)	82 (60.29%)	
Unemployed	15 (8.47%)	13 (9.56%)	
Other	53 (29.94%)	41 (30.15%)	
Annual household income			0.34
<USD 20,000	88 (49.72%)	75 (55.15%)	
≥USD 20,000	89 (50.28%)	61 (44.85%)	
Health Insurance			0.006
No	32 (18.08%)	10 (7.35%)	
Yes	145 (81.92%)	126 (92.65%)	
English proficiency			0.08
Not at all or not well	115 (64.97%)	101 (74.26%)	
Well or very well	62 (35.03%)	35 (25.74%)	

**Table 2 brainsci-12-00188-t002:** Participant Lifestyle Behaviors, Depression, and HBV History by Medication Status (*n* = 329).

	Not on Medication (*n* = 177)	On Medication (*n* = 136)	*p*-Value
Characteristics	*n* (%) or mean (sd)	*n* (%) or mean (sd)	
Smoking status			0.27
Non-smoker	166 (93.79%)	123 (90.44%)	
Smoker	11 (6.21%)	13 (9.56%)	
Drinking status			0.64
Non-drinker	153 (86.44%)	120 (88.24%)	
Drinker	24 (13.56%)	16 (11.76%)	
Physical activity status			0.70
Not at all active	57 (32.20%)	41 (30.15%)	
Physically active	120 (67.80%)	95 (69.85%)	
Depression severity			0.71
No or minimal depression	103 (58.19%)	82 (60.29%)	
Mild to severe depression	74 (41.81%)	54 (39.71%)	
Years since HBV diagnosis	21.72 (11.78)	19.08 (11.17)	0.046

**Table 3 brainsci-12-00188-t003:** Multivariable Logistic Regression Results by Medication Status.

	Not on Medication (*n* = 177)	On Medication (*n* = 136)
Factor	OR (95% CI)	OR (95% CI)
Chinese ethnicity (ref: Vietnamese)	8.02 (2.21–29.13) *	7.12 (1.80–28.16) **
Female gender (ref: male)	1.68 (0.74–3.81)	1.47 (0.61–3.59)
Age	1.07 (1.03–1.11) **	0.99 (0.95–1.04)
Currently married (ref: not)	0.30 (0.11–0.80) *	0.93 (0.27–3.23)
Education	1.06 (0.42–2.67)	1.94 (0.51–7.35)
Employment status (ref: employed)		
Unemployed	3.47 (0078–15.51)	4.53 (1.06–19.30) *
Not in labor force	0.93 (0.28–3.08)	3.44 (1.04–11.34) *
Income ≥USD 20,000 (ref: <USD 20,000)	1.43 (0.49–4.22)	1.82 (0.72–4.65)
Having health insurance	1.08 (0.41–2.84)	1.59 (0.33–7.60)
Well/very well English proficiency (ref: not at all/not well)	0.83 (0.31–2.24)	0.53 (0.15–1.81)
Years since HBV diagnosis	1.01 (0.97–1.04)	1.00 (0.96–1.04)
Physically active (ref: not at all active)	0.40 (0.17–0.92) *	0.29 (0.12–0.75) **
Smoker (ref: non-smoker)	0.54 (0.11–2.69)	0.47 (0.09–2.41)
Drinker (ref: non-drinker)	1.61 (0.53–4.93)	1.54 (0.44–5.34)
Constant	0.005	0.07
Pseudo R-square	0.18	0.18
Likelihood-ratio chi-squared (df)	44.10 (15) ***	33.75 (15) **

Abbreviations: OR = Odds Ratio, CI = Confidence Interval. * <0.05, ** <0.01, *** <0.001.

## Data Availability

In this section, please provide details regarding where data supporting reported results can be found, including links to publicly archived datasets analyzed or generated during the study. Please refer to suggested Data Availability Statements in section “MDPI Research Data Policies” at https://www.mdpi.com/ethics (21 December 2021). You might choose to exclude this statement if the study did not report any data.

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
