# Peer review of "The Association between Modifiable Lifestyle Behaviors and Depression among Asian Americans with Chronic Hepatitis B by Medication Status"

_brainsci, 2022, doi:10.3390/brainsci12020188_

Round 1

Reviewer 1 Report

The manuscript by Zhu et al. is an interesting, clearly-written, important contribution to the literature.  The aim of this manuscript is to investigate a putative relationship between depression and lifestyle behaviors in Asian Americans with chronic hepatitis B.  The subjects were divided into two groups who were either taking antiviral medication or not.  These two groups were compared on demographic variables as well as lifestyle behaviors.  Furthermore, Zhu et al. assessed the impact of those variables on depression.  There were significant differences between the medicated and non-medicated groups on gender, education, and health insurance coverage.  For the medicated group, there were more male, lower educated, and health insurance covered participants.  The two groups did not differ on modifiable lifestyle behaviors or depression.  However, both Chinese ethnicity (versus Vietnamese) was associated with higher odds of mild to severe depression while engagement in physical activity (versus none) was associated with lower odds for both medicated and non-medicated groups, although this was cross-sectional data.

There were some areas that were unclear.  While it is both important and interesting that the impact of some of the demographic and lifestyle behaviors differed by medication status, it is not clear how well the medicated group is adhering to their prescribed antiviral regime.  If there is data regarding adherence, it would be good to include. If not, it should be addressed.  Perhaps more importantly, clarification of why the physical activity variable was used the way it was.  It seems that the authors had data regarding the number of hours of activity, and possibly the type of activity, that subjects engaged in.  The types of activity (classified as light vs moderate or vigorous) might be impactful as well as the amount of time that subjects engaged in these activities.  Another weakness is the lack of data regarding antidepressant use but the authors addressed that in the discussion.

Author Response

Reviewer 1:

Comments

Responses

It is not clear how well the medicated group is adhering to their prescribed antiviral regime. If there is data regarding adherence, it would be good to include. If not, it should be addressed

We have the data on the medication adherence which we presented in another paper. Since it would not be the focus of this paper, we addressed this briefly in the limitation section on Line 250 – 251.

Clarification of why the physical activity variable was used the way it was.  It seems that the authors had data regarding the number of hours of activity, and possibly the type of activity, that subjects engaged in.  The types of activity (classified as light vs moderate or vigorous) might be impactful as well as the amount of time that subjects engaged in these activities. 

We appreciate the comment from the reviewer. Our sample has a very skewed level of physical activities, with almost all participants engaging in a suboptimal level of physical activity, if at all. Therefore, we used the categories we did.

Reviewer 2 Report

The authors report an analysis of the baseline data of an ongoing randomized controlled clinical trial, which recruited 382 Asian Americans with CHB living in the Greater Philadelphia Area and New York City. This study aimed to investigate the association between depression and modifiable lifestyle behaviors among Asian Americans with CHB. A total of 313 participants of this trial were included in this analysis. The authors showed that different ethnicity, physical activity statuses, employment statuses, ages, and marital statuses were associated with varying degrees of depressive symptoms in this sample of CHB patients.

There are some comments.

Comments:

  1. Results and/or Discussion: According to Methods on page 3 (Line 100 and 108), the author included 313 individuals participating in a trial and excluded 69 from the current analysis. What were the baseline characteristics of these excluded participants? Did these excluded participants exhibit baseline characteristics different from those who entered the study? A more detailed description in Results and/or Discussion is suggested.
  2. Results (Table 3): It is recommended to present odds ratios and P values (rather than n (%)) of each factor.
  3. Discussion: This study recruited individuals living in one geographic area of the US. Are Asian Americans living in the Greater Philadelphia Area and New York City similar to (or different from) those living in other areas? Would the differences, if any, affect the associations observed and limit the generalizability of the results? A discussion is recommended.

Author Response

Reviewer 2:

Comments

Responses

Results and/or Discussion: According to Methods on page 3 (Line 100 and 108), the author included 313 individuals participating in a trial and excluded 69 from the current analysis. What were the baseline characteristics of these excluded participants? Did these excluded participants exhibit baseline characteristics different from those who entered the study? A more detailed description in Results and/or Discussion is suggested.

We conducted further analysis to compare the sociodemographics in Table 1 between the missing cases versus the study analysis sample. The two groups differ only on average age: the missing cases has a slightly younger average age (50.19) than the non-missing cases (53.72). We added this information on Line 109 to 112.

Results (Table 3): It is recommended to present odds ratios and P values (rather than n (%)) of each factor.

We revised Table 3. It was an editing issue. We included the odds ratio

Discussion: This study recruited individuals living in one geographic area of the US. Are Asian Americans living in the Greater Philadelphia Area and New York City similar to (or different from) those living in other areas? Would the differences, if any, affect the associations observed and limit the generalizability of the results? A discussion is recommended.

They do not differ significantly on any sociodemographics. We added this on Line 112 to 113.

Reviewer 3 Report

The analyses and tables need major revisions

Author Response

Reviewer 3:

Comments

Responses

The analyses and tables need major revisions

We have corrected Table 3 and made other revisions based on the comments we received.

Must be improved: methods description

Must be improved: result presentation

Round 2

Reviewer 2 Report

The authors have addressed all the concerns in this version of the manuscript.